# The Effect of Dietary Humic Substances on Cellular Immunity and Blood Characteristics in Piglets

**Lukáš Bujňák [1]**, **Alena Hreško Šamudovská [1]**, **Dagmar Mudroňová [2]**, **Pavel Naď [1]**, **Slavomír Marcinčák [3]**, **Iveta Maskaľová [1]**, **Michaela Harčárová [1]**, **Viera Karaffová [4],\*** and **Martin Bartkovský [3]**

[1] Department of Animal Nutrition and Husbandry, University of Veterinary Medicine and Pharmacy in Košice, Komenského 73, 041 81 Košice, Slovakia

[2] Department of Microbiology and Immunology, University of Veterinary Medicine and Pharmacy in Košice, Komenského 73, 041 81 Košice, Slovakia

[3] Department of Food Hygiene, Technology and Safety, University of Veterinary Medicine and Pharmacy in Košice, Komenského 73, 041 81 Košice, Slovakia

[4] Department of Morphological Disciplines, University of Veterinary Medicine and Pharmacy in Košice, Komenského 73, 041 81 Košice, Slovakia

**\*** Correspondence: viera.karaffova@uvlf.sk; Tel.: +421-905-871-840

**Abstract:** This study's objective was to determine the impact of dietary humic substances on immune response and blood profiles in piglets. A total of 24 crossbred piglets (Slovakian White × Landrace; 35 days old; average body weight of 11.67 kg) were allotted to two dietary groups with (experimental; 5 g·kg$^{-1}$) or without (control; 0 g·kg$^{-1}$) natural humic substances supplementation. In this study, we observed a significant increase of the proportion of CD4+CD8- lymphocytes ($p < 0.001$) in the experimental group. The results also showed a tendency for an increase of the phagocytic activity and the engulfing capacity of phagocytes and the numbers of the other monitored lymphocyte subpopulations (CD3+, CD21+, CD4-D8+, CD4+CD8+, CD4+CD25+) in piglets in the experimental group compared to the control group. Supplementation of humic substances increased serum alkaline phosphatase compared to the control group ($p < 0.05$). Other monitored blood parameters were not significantly affected by dietary treatment. It concluded that inclusion of humic substances in the diet of piglets could have a stimulating effect on cellular immunity, without a negative effect on haematological and biochemical parameters.

**Keywords:** humic substances; phagocytosis; lymphocyte; blood biochemistry; haematology; piglets

## 1. Introduction

Humic substances (HS) are a class of non-nutritive natural organic bioactive compounds formed in soils [1]. They are important humus components whose primary function is to transfer nutrients from the soil to living organisms. HS specifically include humic acid, fulvic acid, and humin as their principal constituents [2]. These components are essential for plant growth [3] and can promote the efficient utilization of nutrients by the plant [4].

Due to the prohibition on the use of antibiotics as growth promoters in the European Union, interest in alternative feed additives for animal production has increased [5]. In the last two decades, the interest in the use of HS in animal nutrition has increased [2,6]. Many authors observed an improvement in the production parameters after the addition of HS into feedstuff during their studies. In recent years, it has been shown that HS added to the feed of monogastric animals such as swine, poultry, and rabbits promotes growth [7–10].

Furthermore, HS have been used as immunostimulatory, antidiarrheal, analgesic, and antimicrobial agents in veterinary practices [11–13]. The addition of biologically active supplements of humic nature to the diets of animals stimulated metabolic processes and the digestibility of nutrients, and also activated the absorption of some mineral elements [14]. Other studies found that HS help to reduce ammonia excretion from manure and improves

the relative number of blood lymphocytes. The modifications mentioned above help the animals' immunity [2,8,15].

However, due to the different sources and types of HS preparations, as well as the fact that there is no single standard for measuring genuine HS effects, their bio-effect depends on specification. Based on our previous results and experiences from poultry studies (broilers and laying hens) on the effect of humic substances (in concentrations of 0.8% and 0.5%, respectively) on the immune status and immune response, we decided to verify this effect in piglets. We observed a significant increase of phagocyte activity and B cell response, as well as a significant increase of CD4+:CD8+ lymphocyte ratio [16,17]. In general, supplementation as a feed additive in pigs has not been well reported compared to poultry, and scientific studies on the influence of HS supplementation in pigs' diets on immunity and metabolism are still relatively limited. Therefore, this study, as a pilot trial, was carried out to assess the effect of 0.5% HS supplementation on the immune indicators, as well as biochemical and haematological variables in the blood of piglets.

## 2. Materials and Methods

### 2.1. Experimental Design

The experiment was carried out in accordance with the "European Directive on the protection of animals used for scientific purposes" [18]. The animals were housed in accredited stables of the Department of Animal Nutrition and Husbandry at the University of Veterinary Medicine in Košice, Slovakia, under required zoohygienic conditions. During the experiment, the average temperature in the stable was $20.2 \pm 1.5\ ^{\circ}\text{C}$, and the relative humidity was $68.5 \pm 4.8\%$. The trial was approved by the Ethics Commission of the University of Veterinary Medicine and Pharmacy in Košice (protocol no. EKV/2022-11).

A total of 24 crossbred (Slovakian White x Landrace) 35-days-old piglets were divided into two groups ($n = 12$; 50% male and 50% female in both groups). Prior to the start of the experiment, an initial average animal body weight (BW) of $11.68 \pm 1.35$ kg in the control group and $11.65 \pm 1.34$ kg in the experimental group was recorded. The following experimental groups were included in the study: the control group and the experimental group, where the experimental group diet was supplemented with a 0.5% HS supplement. The experiment lasted for 4 weeks. The same diets for both groups were used in the experiment (Table 1). The introduction of the HS supplement into the diet was realized at the expense of barley in the experimental group. The pigs were fed twice per day with complete feed mixtures. Drinking water was available to the animals ad libitum throughout the experimental period. The complete feed mixtures in the experiment were formulated according to the nutritional requirements by Šimeček et al. [19].

The dietary natural HS supplement (HUMAC® Natur AFM; Humac, Ltd., Košice, Slovakia) was ground and physically purified with Leonardite without chemical treatment.

The complete feed mixtures were analysed for dry matter, crude protein, crude fibre, total ash, starch, and total phosphorus according to EC Commission Regulation 152/2009 [20]. The level of dietary calcium and sodium was analysed using the flame method of an atomic absorption spectrometer (Unicam Solar 939, Camberley, Surrey, UK). The metabolisable energy values of complete feed mixtures were calculated with the formula according to the Šimeček et al. [19].

### 2.2. Sampling and Measurements

In week 4, in the morning on the last day of the experiment, blood samples were taken by the *orbital sinus* puncture from all pigs individually in both groups for subsequent analysis. Serum was obtained by centrifugation (3000 rpm for 30 min) and stored at $-20\ ^{\circ}\text{C}$ until analysis. Heparinized blood was used to determine haematological parameters, test phagocyte activity and identify lymphocyte subpopulations.

**Table 1.** Formula and chemical composition of feed mixtures for pigs (as-fed basis).

| | | Control Diet | Experimental Diet |
|---|---|---|---|
| Components | | | |
| Corn | [%] | 25.00 | 25.00 |
| Wheat | [%] | 22.50 | 22.50 |
| Barley | [%] | 28.00 | 27.50 |
| Soybean meal | [%] | 21.00 | 21.00 |
| Vitamin-mineral premix with 3-phytase [1] | [%] | 3.00 | 3.00 |
| Sodium chloride | [%] | 0.12 | 0.12 |
| L-Lysine | [%] | 0.23 | 0.23 |
| DL-Methionine | [%] | 0.06 | 0.06 |
| L-Threonine | [%] | 0.09 | 0.09 |
| HS supplement [2] | [%] | - | 0.50 |
| Composition by analysis | | | |
| Dry matter | [g·kg$^{-1}$] | 885.2 | 884.3 |
| Crude protein | [g·kg$^{-1}$] | 178.2 | 177.4 |
| Crude fibre | [g·kg$^{-1}$] | 35.4 | 38.9 |
| Ash | [g·kg$^{-1}$] | 54.1 | 57.6 |
| Starch | [g·kg$^{-1}$] | 445.0 | 438.0 |
| Calcium | [g·kg$^{-1}$] | 6.4 | 7.0 |
| Phosphorus | [g·kg$^{-1}$] | 5.6 | 5.6 |
| Sodium | [g·kg$^{-1}$] | 2.7 | 2.8 |
| Metabolisable energy | [MJ·kg$^{-1}$] | 13.07 | 13.02 |

[1] Vitamin and mineral premix (per kg): vit. A 330,000 IU; D$_3$ 66,000 IU; E 4000 mg; calcium 210 g; phosphorus 25 g; sodium 45 g; copper 4800 mg; iron 1750 mg; zinc 2700 mg; manganese 880 mg; iodine 55 mg; selenium 12.5 mg. [2] The characteristics of the HS supplement were the following: the size of particles up to 100 μm, pH 5.8, humidity max. 15%, content of humic acids min. 65% in dry matter (DM), fulvic acid 5% (DM).; minerals: calcium 42.28, magnesium 5.11, sodium 7.11, potassium 0.93 g·kg$^{-1}$; and microelements: Fe 19,046; Cu 15; Zn 37; Mn 142; Co 1.24; Se 1.67, as well as Mo 2.7 mg·kg$^{-1}$ DM.

### 2.2.1. Haematological and Serum Biochemical Parameters

Complete blood count was performed with an automated haematology analyser (scil Vet ABC™ Hematology Analyzer, Germany). The variables evaluated in our study were haematocrit value (HCT), haemoglobin concentration (Hb), red blood cells count (RBC), mean corpuscular volume (MCV), and total white blood cell (WBC) count. Serum biochemical parameters—total protein, albumin, glucose, urea, triglycerides, cholesterol, creatinine, aspartate aminotransferase (AST), alkaline phosphatase (ALP), and phosphorus were measured using a fully automatic random access benchtop analyser (Ellipse, Italy). The concentration of calcium in serum was determined by means of a flame atomic absorption spectrometer (Unicam Solar 939, Camberley, Surrey, UK).

### 2.2.2. Biomarker of the Lipid Peroxidation

The concentrations of lipid peroxidation products (malondialdehyde levels, MDA) in serum were measured as thiobarbituric acid reactive substances (TBARs) according to the spectrophotometric modification method described by Costa et al. [21]. Briefly, serum samples were mixed with a solution composed of trichloroacetic acid (15%; Merck, Darmstadt, Germany), thiobarbituric acid (0.38%; Sigma-Aldrich, St. Louis, MO, USA), and hydrochloric acid (0.25 N; Mikrochem, Pezinok, Slovakia) and heated for 30 min in a boiling water bath. After cooling in ice water and centrifugation, the absorbance of the supernatant was measured at 535 nm. The concentration of TBARs was determined from the standard curve prepared using 1,1,3,3-tetramethoxypropane (malondialdehyde-bis (dimethyl acetal); Acros Organics, Geel, Belgium). The results were expressed as nmol of MDA/mL of serum.

### 2.2.3. Phagocyte Activity Testing and Identification of Lymphocyte Subpopulations

A commercial Phagotest® assay (Celonic, Munich, Germany) was used to determine the phagocytic activity and the engulfing capacity of the phagocytes. The manufacturer's instructions were followed when performing the test.

Direct immunostaining assay was applied to identify selected subpopulations of lymphocytes. Two combinations of conjugated mouse anti-porcine monoclonal antibodies: CD3e/CD21 and CD4/CD8a/CD25 were used according to the specifications given in Table 2. Heparinised blood in amount of 50 μL was incubated with monoclonal antibodies for 15 min in the dark at laboratory temperature. After incubation, 1 mL of BD FACS lysis solution was added to the tubes. The tubes were incubated for an additional 20 min in the dark at laboratory temperature and then centrifuged ($300\times g$ for 5 min). Cell pellets were then washed and centrifuged twice with 1 mL phosphate buffer saline (PBS; MP Biomedicals, Illkirch-Graffenstaden, France) at $300\times g$ for 5 min. Finally, cells were resuspended in 200 μL of PBS for subsequent cytometric analysis.

**Table 2.** Specification and amounts of used mouse anti-porcine monoclonal antibodies.

| Type | Fluorochrome | Clone | Amount/50 μL Blood | Producer |
|---|---|---|---|---|
| anti-CD3e | FITC | BB23-8E6 | 4 μL | BD Biosciences, Franklin Lakes, NJ, USA |
| anti-CD4 | FITC | MIL 17 | 4 μL | AbD Serotec, Kidlington, UK |
| anti-CD8a | R-PE | MIL 12 | 2 μL | AbD Serotec, Kidlington, UK |
| anti-CD25 | PE-Cy7 | PC 61.5 | 1 μL | eBioscience, San Diego, CA, USA |
| anti-CD21 | R-PE | BB6-11C9.6 | 2 μL | SuthernBiotech, Homewood, AL, USA |

CD—cluster of differentiation; FITC—fluorescein isothiocyanate; R-PE—R-phycoerythrin; PE-Cy7—phycoerythrin-cyanine 7.

Phagocytic activity analysis as well as the identification of lymphocyte subpopulations was performed on a six-colour BD FACSCanto™ flow cytometer (Becton Dickinson Biosciences, San Jose, CA, USA) using BD FACS Diva™ software. The position of the analysed cells was gated in FSC vs. SSC dot plots. Granulocytes, monocytes, and both cell populations together, respectively, were gated for phagocytic activity analysis. Based on the low DNA content in the red fluorescence histogram (FL-2), bacterial aggregates were excluded from further analysis. The percentage of active phagocytes and the mean fluorescence intensity were determined in the green fluorescence histogram (FL-1).

Gated lymphocytes (Figure 1a,b) were used for the identification of lymphocyte subpopulations. CD3+ lymphocytes represent T lymphocytes and CD21+ B lymphocytes (Figure 1c). The CD4+CD8- subpopulation was evaluated as T helper lymphocytes, CD4-CD8+ as cytotoxic lymphocytes, CD4+CD8+ as double positive T lymphocytesy (Figure 1d), and CD4+CD25+ as regulatory T lymphocytes (Figure 1e). Proportions of lymphocytes are expressed in percentage.

### 2.3. Statistical Analysis

The results were statistically evaluated by an unpaired $t$-test with the statistical software GraphPad Prism 8.0. A value of $p < 0.05$ was considered statistically significant. The results obtained in this experiment were expressed as mean $\pm$ standard error of the means (SEM).

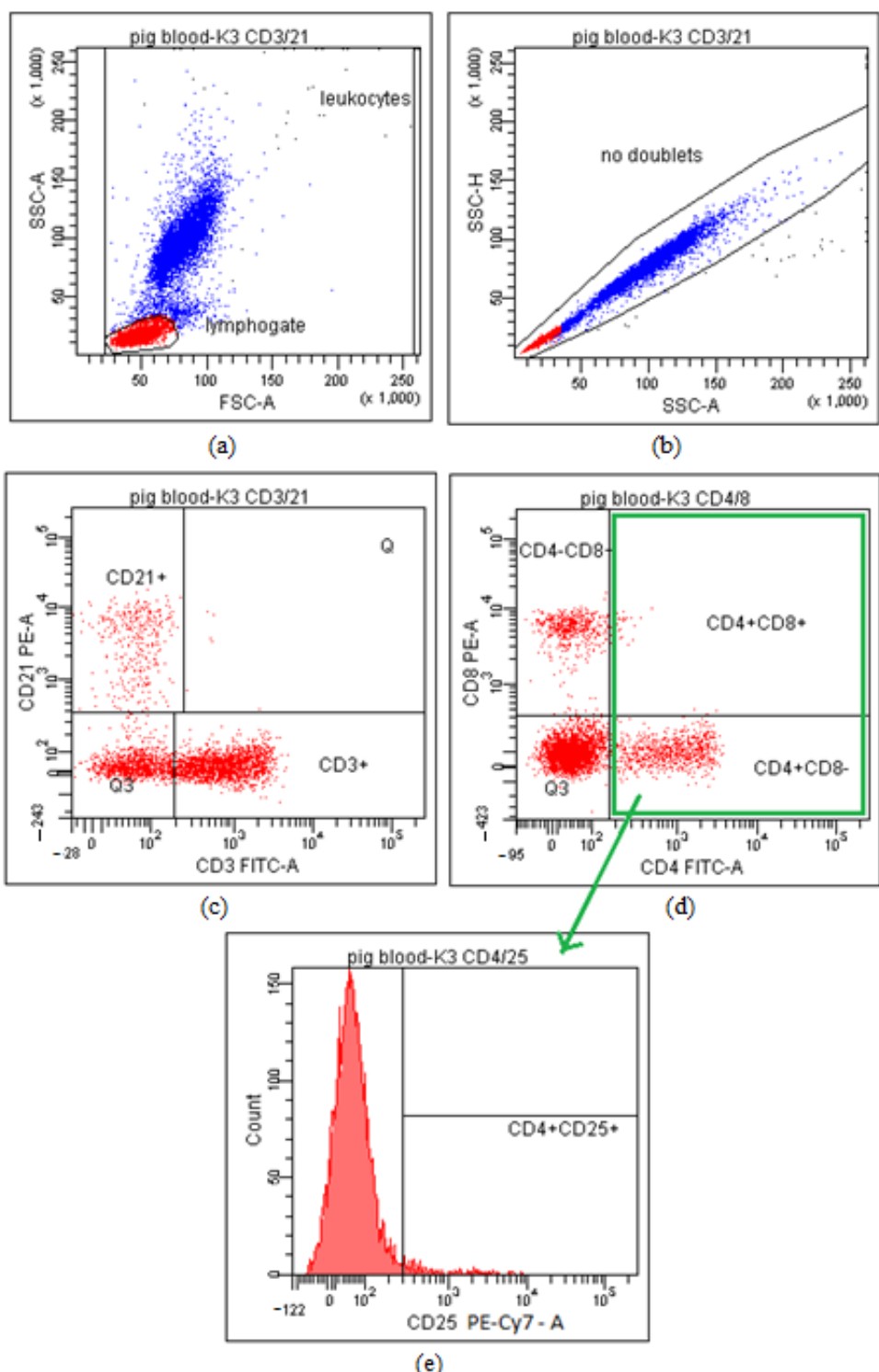

**Figure 1.** Gating strategy: (**a**) determination of the position of lymphocytes (red) on the basic dot plot (FSC-A versus SSC-A); (**b**) removing doublets and agglomerates from the analysis; (**c**) analysis of the representation of CD3+ and CD21+ lymphocytes; (**d**) analysis of the representation of CD4+ and CD8+ lymphocytes; (**e**) determination of the representation of CD4+CD25+ lymphocytes on histogram from CD4+ positive cells (gated with green).

## 3. Results

### 3.1. Haematological and Serum Biochemical Parameters

Effects of HS on haematological and biochemical blood profile are presented in Table 3. The determined blood parameters including content of red blood cells, white blood cells, haemoglobin, haematocrit, mean corpuscular volume, total protein, albumin, glucose, urea, triglycerides, cholesterol, creatinine, AST, Ca, and P were unaffected by the dietary HS treatment. However, dietary supplementation with HS increased activity of ALP ($p < 0.05$).

**Table 3.** Effect of humic substances on some haematological and serum biochemical parameters in piglets.

|  |  | Control | Experimental | *p*-Value |
|---|---|---|---|---|
| Haematological indices |  |  |  |  |
| Red blood cells | [T·L$^{-1}$] | 6.82 ± 0.10 | 6.81 ± 0.10 | 0.954 |
| Mean corpuscular volume | [fL] | 51.00 ± 0.26 | 52.00 ± 0.89 | 0.308 |
| Haematocrit | [L·L$^{-1}$] | 0.35 ± 0.01 | 0.35 ± 0.01 | 0.739 |
| Haemoglobin | [g·dL$^{-1}$] | 11.68 ± 0.25 | 11.58 ± 0.19 | 0.755 |
| White blood cells | [G·L$^{-1}$] | 10.87 ± 0.20 | 12.28 ± 1.24 | 0.287 |
| Serum metabolites |  |  |  |  |
| Total protein | [g·L$^{-1}$] | 60.03 ± 0.55 | 60.41 ± 0.68 | 0.675 |
| Albumin | [g·L$^{-1}$] | 28.78 ± 1.76 | 32.80 ± 0.87 | 0.067 |
| Glucose | [mmol·L$^{-1}$] | 5.05 ± 0.05 | 5.19 ± 0.10 | 0.217 |
| Urea | [mmol·L$^{-1}$] | 3.89 ± 0.20 | 3.73 ± 0.25 | 0.621 |
| Triglycerides | [mmol·L$^{-1}$] | 0.78 ± 0.09 | 0.89 ± 0.07 | 0.375 |
| Cholesterol | [mmol·L$^{-1}$] | 2.45 ± 0.17 | 2.47 ± 0.19 | 0.934 |
| Creatinine | [μmol·L$^{-1}$] | 79.90 ± 3.55 | 81.10 ± 3.72 | 0.820 |
| ALP | [μkat·L$^{-1}$] | 4.24 ± 0.12 | 4.60 ± 0.10 * | 0.041 |
| AST | [μkat·L$^{-1}$] | 0.84 ± 0.03 | 0.88 ± 0.06 | 0.546 |
| Ca | [mmol·L$^{-1}$] | 2.68 ± 0.04 | 2.82 ± 0.05 | 0.052 |
| P | [mmol·L$^{-1}$] | 2.09 ± 0.02 | 2.11 ± 0.04 | 0.692 |
| TBARs | [nmol MDA·mL$^{-1}$] | 0.72 ± 0.03 | 0.69 ± 0.03 | 0.562 |

ALP—alkaline phosphatase, AST—aspartate aminotransferase, TBARs—thiobarbituric acid reactive substances. * Column labelled with asterisk is significantly different from the control ($p < 0.05$).

Serum TBARs level, secondary products of lipid peroxidation (which are an important parameter used in the determination of lipid peroxidation), was not affected by the addition of HS.

### 3.2. Cellular Immune Response

The addition of HS to piglets' feed did not have a significant effect on the percentage of active phagocytes (Table 4) and the engulfing capacity of the phagocytes (Table 5). The higher values of these innate immune response parameters were found in the experimental group compared to the control group.

**Table 4.** Effect of humic substances on the phagocyte activity in piglets' blood evaluated as percentage of active phagocytes—phagocytic activity.

|  | PA$_{total}$ [%] | PA$_{Neu}$ [%] | PA$_{Mo}$ [%] |
|---|---|---|---|
| Control | 83.08 ± 0.62 | 85.68 ± 0.60 | 77.18 ± 1.76 |
| Experimental | 84.90 ± 1.33 | 88.04 ± 0.94 | 78.08 ± 2.10 |
| *p*-Value | 0.251 | 0.067 | 0.751 |

PA$_{total}$—total phagocytic activity, PA$_{Neu}$—phagocytic activity of neutrophils, PA$_{Mo}$—phagocytic activity of monocytes.

**Table 5.** Effect of humic substances on the engulfing capacity of the phagocytes expressed as mean fluorescence intensity (MFI).

|  | $MFI_{total}$ | $MFI_{Neu}$ | $MFI_{Mo}$ |
|---|---|---|---|
| Control | $31{,}775 \pm 3010$ | $34{,}783 \pm 3223$ | $18{,}744 \pm 1858$ |
| Experimental | $36{,}057 \pm 1097$ | $39{,}560 \pm 1220$ | $19{,}041 \pm 929$ |
| *p*-Value | 0.218 | 0.203 | 0.889 |

$MFI_{total}$—total mean fluorescence intensity, $MFI_{Neu}$—mean fluorescence intensity of neutrophils, $MFI_{Mo}$—mean fluorescence intensity of monocytes.

The percentages of selected lymphocyte subpopulations are presented in Figure 2a–f. It was found that the addition of HS to the diets of piglets significantly increased the proportion of T helper cells (CD4+CD8-) ($p < 0.001$). The results also showed a tendency towards an increase in the number of other monitored lymphocyte subpopulations (CD3+, CD21+, CD4-CD8+, CD4+CD8+, CD4+CD25+) in the piglets in the experimental group compared to the control group, but without a significant difference. Additionally, the ratio of CD4+ and CD8+ lymphocytes (Figure 2g), as a marker of immune stimulation, showed a non-significant increase.

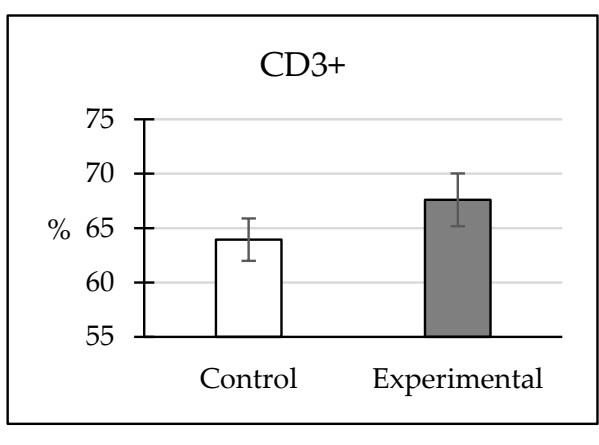

(**a**)

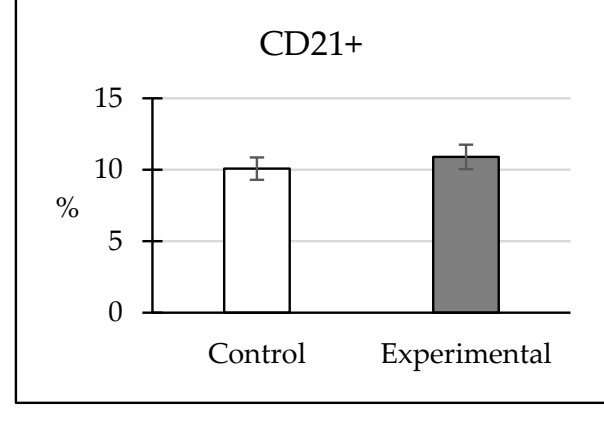

(**b**)

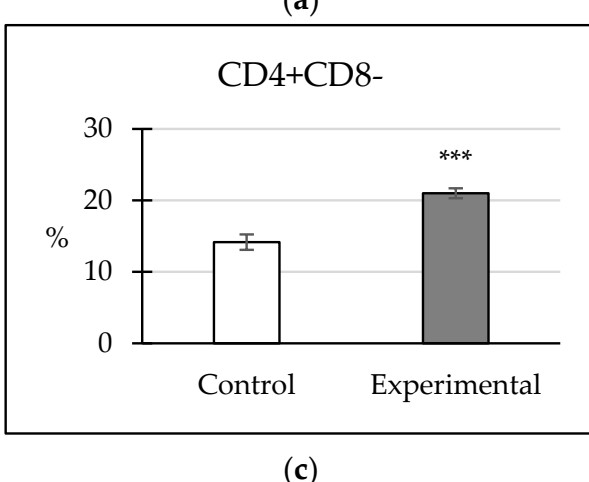

(**c**)

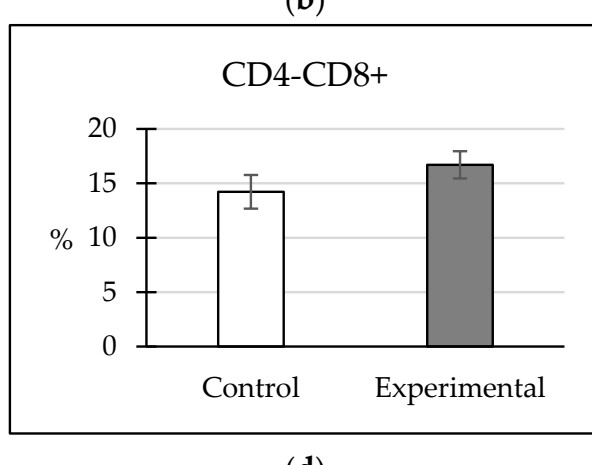

(**d**)

**Figure 2.** *Cont*.

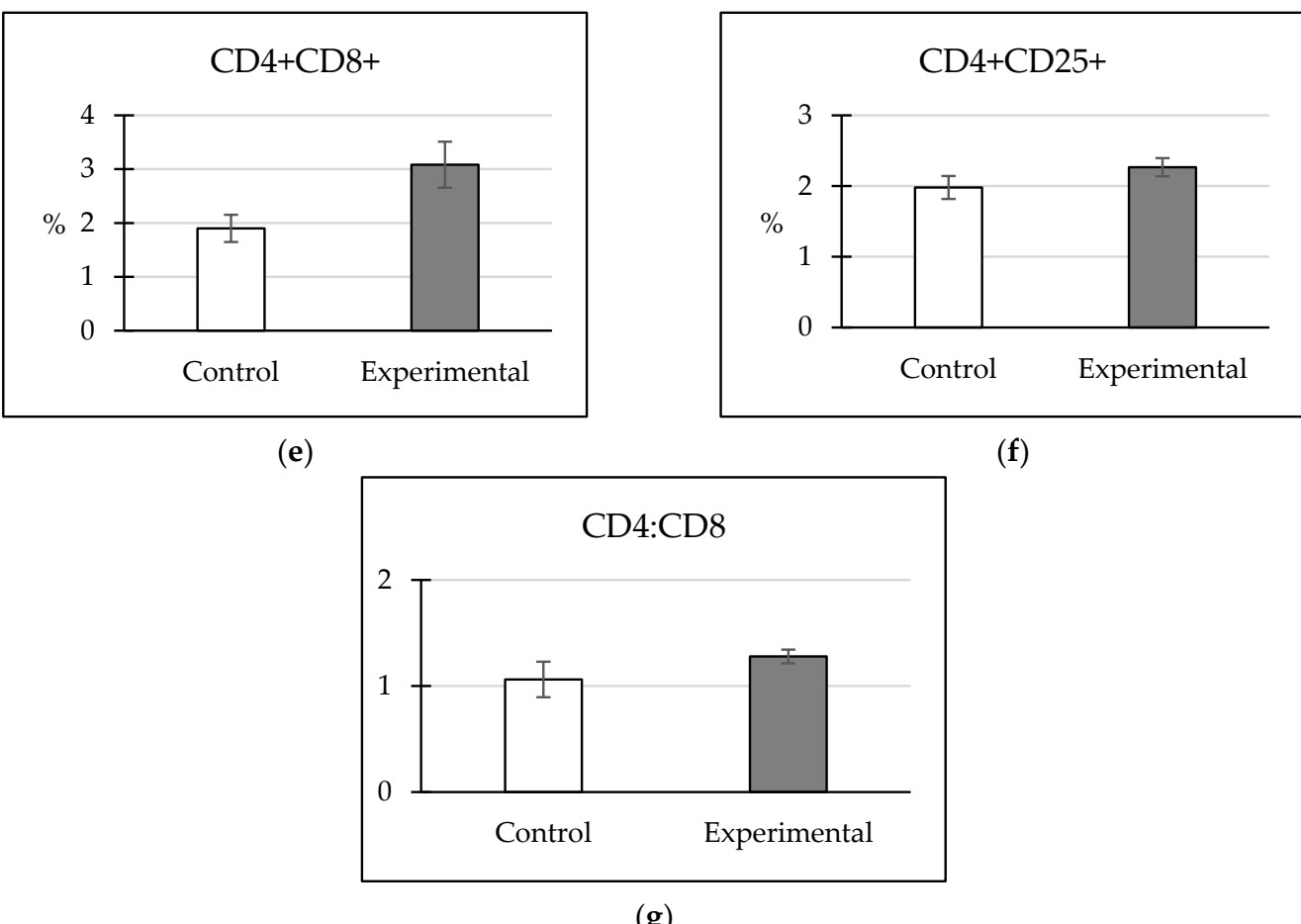

**Figure 2.** Effect of humic substances on the percentage of lymphocytes: (**a**) CD3+; (**b**) CD21+; (**c**) CD4+CD8-; (**d**) CD4-CD8+; (**e**) CD4+CD8+; (**f**) CD4+CD25+; and (**g**) ratio of CD4:CD8 lymphocytes in the blood of the piglets. Column labelled with asterisk is significantly different from the control (*** $p < 0.001$).

## 4. Discussion

### 4.1. Cellular Immune Response

Our study's major objective was to track how humic substances (HS) affected a few different cellular immunity indicators. The percentage of active phagocytes and their engulfing capacity were selected as indicators for monitoring the effect on innate cellular immunity. In our previous study, it was found that the addition of HS to the diets (0.8%) of broilers significantly increased phagocytic activity as well as mean fluorescence intensity [16]. Additionally, ELnaggar and El-Kelawy [22] observed an increase in phagocytic activity and phagocytic index after humic acid supplementation to the diet (0.1, 0.2, and 0.4%) of Sacco chickens. Similar findings were obtained with laying hens [17]. In the current study, the supplementation of HS to the diet of piglets had no significant effect on phagocytosis. However, the percentage of active phagocytes as well as their engulfing capacity were numerically higher in piglets supplemented with HS than in piglets in the control group. According to Sanmiguel and Rondón [23], the effect of HS on phagocytes is time-dependent. They found that the supplementation of HS to the diets (0.1 and 0.2%) of laying hens increased the phagotic index on day 8 and 30 of application, but on day 60, it was lower than the control group.

The representation of the selected lymphocyte subpopulations in the blood was selected as a parameter for monitoring the effect of HS on the acquired cellular immunity. Results of the present study showed a significant increase in T helper lymphocytes (CD4+CD8-) when 0.5% HS were added to the diet of piglets. The outcomes can be partially

compared to those of Wang et al. [2], who found that pigs whose feed had 10% HS added to it, had higher relative lymphocyte counts. Similar results were obtained for poultry, which were fed a diet supplemented with 0.15% humate [24], or were supplemented with 20 mg humic acid/kg of body weight in drinking water [25].

According to Cetin et al. [24], the supplementation of laying hens' diets with humic compounds results in a significant increase in the lymphocyte counts, which can be attributed to increased production of IL-2, as well as expression of the IL-2 receptor on lymphocytes. Humic substances seem to enhance the activity of IL-2 producing cells.

However, other authors did not observe any significant effect on lymphocyte count in pigs that were fed with humic acid supplemented feed [26] as well as in chickens that were given humic acids in their drinking water [27] or their diets [22].

Our results are similar with the observations of Mudroňová et al. [16], who found that the percentage of CD4 lymphocytes was significantly increased for broilers fed HS supplementation (0.8%) compared with the control group. They also noted a significant decrease in CD8+ lymphocytes (T cytotoxic lymphocytes), which resulted in a statistically higher CD4:CD8 ratio, which is used as a marker of immune stimulation. On the other hand, feeding laying hens with a diet of 0.5% HS significantly increased the proportion of IgM+ lymphocytes that represent a subpopulation of B lymphocytes and significantly reduced the proportion of CD3+ lymphocytes that represent total T lymphocytes. The proportion of T helper and T cytotoxic lymphocytes was not affected [17].

Based on our results, as well as various studies, it follows that HS can have an immunostimulatory effect which can be influenced by the composition and quantity of humates used, the method of their administration, the animal species, and, according to ELnaggar and El-Kelawy [22], by the rearing of animals in various regions of the world, differing in climate. The immunomodulation of HS may theoretically consist of the formation of complexes of humates with saccharides. These complexes bind to the surface of T lymphocytes and NK cells and affect their function, including the production of cytokines, which further influence other cells of the immune system [28].

### 4.2. Haematological and Serum Biochemical Parameters

Another objective of this study was to evaluate selected haematological and biochemical blood parameters of piglets fed diets supplemented with HS. No significant difference in selected variables of protein, energy, and mineral metabolism, as well as in haematological indices, was found between the experimental group of animals with dietary HS supplement and the control group, except for ALP activity in the current study. Biochemical indicators in our study were within the reference range in all piglets, according to Doubek et al. [29] and Kraft and Dürr [30].

Our findings are in line with prior work by Wang et al. [2], who found no significant alterations in the red and white blood cells during the course of the trial in their study to find out the effects of HS on blood variables in finishing pigs.

After treatment with an HS-containing diet, Herzig et al. [31], and also Šamudovská and Demeterová [32], observed non-significantly increased ALP activity in chickens. In our investigation, we found that the experimental group had significantly higher ALP activity than the control group. Their values, meanwhile, remained within the reference range for pig ALP levels (2–17 $kat \cdot L^{-1}$) [29,30].

On the other hand, Jad'uttová et al. [33] and Rath et al. [34] noticed a significant reduction in ALP activity in chickens following feeding with HS. The blood levels of ALP in broilers given a 1.00% HS feed supplement decreased, according to these authors. Although the decreased values of ALP activity in the Rath et al. [34] study were statistically different than the controls, they did not reflect any toxic effect of HS on selective organs (muscle, kidney, heart, or liver).

The amount of HS provided by the diet can affect the concentration of minerals in the blood. The levels of calcium and phosphorus in the piglets' blood serum were unaffected by the administration of HS at a concentration of 0.5%. The ability of HS to

chelate metals, which is influenced by a significant amount of carboxylic acid side-chains, could be the reason for the decrease in mineral serum concentrations [34]. In our study, the concentrations of calcium and phosphorus in piglets' blood were not decreased below reference values by the addition of HS to the feed mixture of the experimental group.

MDA is the product of lipid peroxidation caused by oxygen free radicals, which can be used as a marker to assess antioxidant status and lipid peroxidation [35]. MDA was determined as a biomarker of oxidative stress by measuring the levels of thiobarbituric acid reactive substances (TBARs) in the blood. A decrease in oxidative stress is characterized by a reduction in MDA levels.

Previous research has shown that HS may have antioxidant properties, protecting against a variety of disorders connected to the oxidative stress that free radicals typically cause. For instance, Wang et al. [15] found that dietary supplementation of sodium humate (2000 mg·kg$^{-1}$) could improve the antioxidant status of weaned piglets. They observed the significant reducing content of MDA in the serum, as well as a significant increase in total antioxidant capacity. However, HS are natural materials with a variable composition, which can cause different effectiveness, which can also depend on their administered amount. In the current experiment, no variation was observed in serum MDA (TBARs) levels of piglets fed HS in comparison to the control group. Our results are consistent with Zhang et al. [36], who found that the addition of sodium humate to the diets (0.1, 0.3, and 0.5%) of laying hens had no effect on serum total antioxidant capacity and MDA values.

## 5. Conclusions

Based on the results obtained in this study, it may be concluded that a diet supplementation with 0.5% humic substances could have a stimulatory effect on some immune cells in piglets. There was a significant increase in the proportion of CD4+CD8- lymphocytes in the blood. Supplementation with humic substances increased serum ALP. However, these values were still within the reference range. The results showed that even such a low concentration of HS can positively affect cellular immunity in piglets. Due to the fact, that the concentration of humic substances we chose had a significant effect only on some indicators of cellular immunity, further studies will be necessary to choose an appropriate concentration and confirm the stimulating effect on immunity in this category of animals.

**Author Contributions:** Conceptualization, L.B., A.H.Š., D.M., V.K. and S.M.; methodology, L.B., A.H.Š., I.M. and D.M.; formal analysis, L.B. and A.H.Š.; investigation, L.B., M.H. and M.B.; data curation, A.H.Š., L.B. and D.M.; writing—original draft preparation L.B. and A.H.Š.; writing—review and editing, L.B., A.H.Š., D.M. and S.M.; visualization, L.B. and A.H.Š.; supervision, P.N.; project administration, L.B. and A.H.Š.; funding acquisition, P.N. and S.M. All authors have read and agreed to the published version of the manuscript.

**Funding:** This publication was supported by the Operational Program Integrated Infrastructure within the project: Demand-Driven Research for Sustainable and Innovative Food, Drive4SIFood 313011V336, co-financed by the European Regional Development Fund (50%) and by the Ministry of Education, Science, Research and Sport of the Slovak Republic (Project VEGA No. 1/0402/20 "Effect of additives in nutrition of monogastric animals on production health, production parameters, product quality and the environment") (50%).

**Institutional Review Board Statement:** All procedures in the present study were performed in accordance with the principles of the European Union and Slovak Law on Animal Protection.

**Data Availability Statement:** The data presented in this study are available upon request from the corresponding author.

**Acknowledgments:** The authors gratefully acknowledge Humac, Ltd., Košice, Slovakia, for providing humic substances for the experiment.

**Conflicts of Interest:** The authors declare no conflict of interest.

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
