# Peer review of "The Effect of Dietary Humic Substances on Cellular Immunity and Blood Characteristics in Piglets"

_agriculture, doi:10.3390/agriculture13030636_

Round 1

Reviewer 1 Report

See that attached

Author Response

Communication: The effect of dietary humic substances on cellular immunity and blood characteristics in piglets

Manuscript ID: 2251264

Authors:  Bujňák et al.

Dear Reviewer,

Firstly, thank you for your time, willingness, and valuable comments, which helped us improve the overall quality of the revised manuscript. All accepted changes and modifications from all reviewers were marked up using "Track Changes" (MS Word) in the revised manuscript.

Yours sincerely,

team of authors

General comments

Reviewer 1

The paper needs to be revised for English grammar and sentence structure by a native English speaker.

  • We had our manuscript checked by a native English-speaking person from Ireland. If it is needed, we are prepared to use a paid editing service (Language Editing Services; https://www.mdpi.com/authors/english).

There is lack of proper justification for HS supplementation at 0.5%. This seems too low which would explain the lack of response in majority of the data, and questions the conclusions.

  • We understand your note about concentration level (0.5%) of HS in the study.

Allow us to clarify this:

  • Since this is a pilot study to assess the effect of 0.5% HS supplementation on immune parameters as well as biochemical and haematological variables in piglet blood, we prepared our manuscript as a communication or short note. 
  • Based on our previous results and experiences from poultry studies on the effect of humic substances on the immune status and immune response, we decided to verify this effect in piglets.
    • after the application of 0.5% HS to laying hens, we observed a significant increase in both the percentage of active phagocytes and their engulfing capacity - Mudroňová, D.; Karaffová, V.; Semjon, B.; Naď, P.; Koščová, J.; Bartkovský, M.; Makiš, A.; Bujňák, L.; Nagy, J.; Mojžišová, J.; Marcinčák, S. Effects of dietary supplementation of humic substances on production parameters, immune status and gut microbiota of laying hens. Agriculture, 202111, 744. DOI: 10.3390/agriculture11080744
  • We respect your note that this level is only similar to trials with poultry, not pigs, however some authors reported that addition of humic substances in lower concentrations can have significant effect:
    • the beneficial effect of feeding humates in a 0.25% concentration during broiler fattening was observed in Kocabagli et al. (2002) - Kocabağli, N.; Alp, M.; Acar, N.; Kahraman, R. The effects of dietary humate supplementation on broiler growth and carcass yield.  Sci.200281, 227–230. 
    • Weber et al. (2014) suggested that HS in the diet of young pigs (0.25% for 35 days) may play a role in negating the effects of oxidative stress within the body. - Weber TE et al.(2014): Effects of dietary humic and butyric acid on growth performance and response to lipopolysaccharide in young pigs. Journal of Animal Science 92, 4172–4179.

  • It is also true that supplementation of HS as a feed additive in pigs has not been well reported compared to poultry, and scientific studies on the influence of humic substances supplementation in pigs' diets on growth intensity, metabolism, and immunity are still relatively limited. Sometimes it is difficult to compare the actual effects of HS preparations due to their different sources and nature, as well as because their bio-effect depends on specification.
    • Articles of Wang et al. (2008) – 10%HS and also Kim et al. (2019) – 2%HS were realised with finishing pigs.
  • At the moment, we are testing other concentrations and other properties (e.g. particle size) of the humic preparation in piglets.

The materials and method section as written is quite inconsistent, difficult to comprehend or missing some vital information and needs to be revised for clarity (See specific comments).

  • The section has been modified in the manuscript (see answers on specific comments).

There is inconsistent use of piglets, weanling pigs, young pigs to mean the same thing, the authors should stick to one.

  •             It has been modified in the manuscript (as "piglets").

Specific comments:

Reviewer 1

LN 18: Revise as “…humic substances on immune response in piglets”. What’s the age/BW of the pigs?

LN 20: “…allotted to two dietary groups with (5 g/kg) or without (Control; 0 g/kg) HS supplementation.

LN 22: What treatment group is being referred to here?

LN 23: Delete “non-significant”

LN 26: Define ALP

LN 34: Add citation after “soils”

LN 37 – 41: Two different biological systems so not properly justified

LN 44: Add citation after “increased”

LN 49: “Furthermore, HS have been used as an immuno…”

LN 51: Not all of these citations discuss HS use

LN 59: Describe some of the results from the Poultry studies

LN 64: What informed the choice of 0.5%? Was this based on a pilot trial? Literature? Chemical composition?

LN 76: What is the BW of the pigs?

  • We accept all of this and they have been changed and modified in the manuscript.

LN 80 – 81: The same diets for both groups? What percentage of the body weight were the pigs fed at?

  • The same diets for both groups were used in the experiment (Table 1). The introduction of the HS supplement into the diet was realized at the expense of barley in the experimental group. The pigs were fed twice per day with complete feed mixtures and were allowed ad libitum access to water throughout the experiment. The diets were formulated to meet or exceed the nutritional requirements recommended of the Šimeček et al. [19].
  • Note: Average daily feed intake was in the 4 week experiment 1.13kg in the both groups.

LN 87 – 88: Include the name of the HS vendor

  • In the Materials and method section

The dietary natural HS supplement (HUMAC® Natur AFM; Humac, Ltd., Košice, Slovakia) was grounded and physically purified Leonardite without chemical treatment.

LN 87 – 95: This information may be better as a footnote description in the diet table

  • It has been changed in the manuscript.

LN 102: delete [%] or reformat to bring it close to the listed items in the table. Also, add analyzed gross energy concentration of the diets to the table, in addition to ME.

  • Table 1. (Formula and chemical composition of feed mixtures for pigs) has been modified as suggested by another reviewer.

LN 106 – 111: How many pigs per rep? Blood collection via orbital sinus is not appropriate for pigs this age (welfare concerns). What informed the choice, and not venipuncture?

  • The blood samples were taken from all pigs individually in both groups (12+12).
  • The orbital sinus collection technique is minimally invasive, and is quick. Pigs exhibit minimal discomfort during the procedure, if sampling is done correctly. We have experience with this method. In older pigs we use venepuncture.

LN 120: What is the method for Ca determination?

  • The concentration of calcium in serum was determined by means of flame atomic absorption spectrometer (Unicam Solar 939, Camberley, Surrey, UK).

Note:

The blood serum was diluted 1:1 with La2O3 in a 50 ml volumetric flask. This solution was analysed for the content of Ca by using a flame atomic absorption spectrometer. The flame conditions were those recommended by the instrument manufacturer for Ca (wavelength 422.7) Determinations were done according to the methodology specified in the List of Official Methods and Laboratory Diagnostics of Food and Feed in the Bulletin of the Ministry of Agriculture the Slovak Republic, 2004.

LN 158 – 170: Not sure this is relevant in-text. Perhaps it could be added as separate supplementary figures.

  • A representative figure of flow cytometry results was added to the manuscript on the recommendation of a past reviewer and also our immunology specialist.

LN 171 – 175: How did the authors account for effect of sex??

  • The influence of sex was not separately evaluated statistically because the values of individual indicators did not differ within the group.

LN 207: HS increased T Helper cell % in response to what? Since there is no change due to HS in hematological/serological data

  • In the hematological profile, we only evaluated the total number of white blood cells; individual populations were not monitored, and therefore the fact that some subpopulations of lymphocytes were increased may not mean an increase in total leukocytes.

LN 218 – 241: Again, why HS at 0.5%? This level is only similar to trials with chickens, not pigs which are bigger and more complex. In Ln 239, pigs were supplemented with 10% HS.

  • As stated in the general comments section, this is a pilot study written as a short note or communication; it is with a young animal category (piglets); and it is with a local commercial dietary natural HS supplement (HUMAC® Natur AFM; Humac, Ltd.; Slovakia). The main goal of this study was to assess the influence of HS on cellular immunity with an assessment of lymphocyte subpopulations.
  • The result is that even such a low concentration of HS can positively affect immunity in piglets - a significant increase of the proportion of CD4+CD8- lymphocytes (P < 0.001) in the experimental group and a tendency for increase of the phagocytic activity and the engulfing capacity of phagocytes and the numbers of the other monitored lymphocyte subpopulations.

LN 323: This is misleading because the lack of effect in majority of the data and previous reports suggests HS was supplemented at a subpar level not likely to elicit any response. Hence the authors cannot conclude that the pigs tolerated it.

  • It has been changed in the manuscript.

Reviewer 2 Report

The study by Bujňák et al. assessed the effects of dietary supplementation of HS (0.5%) on some cellular immune indicators in pigs. The paper is well-written and exciting. Some comments should be considered before further processing

Abstract: please define all abbreviations first mentioned.

Why did the authors use only one supplement level (0.5%)? They should assess different graded levels to investigate the supplementation level that gives the best results finally.

Please avoid using “humic substances” and replace them with the exact supplement used.

Information about the number of replications, rearing, and management of pigs during the experiment should be provided.

L116-121: please mention in brief the methodology of analysis of these parameters.

Please insert the p-value in the tables (results)

The discussion only compares the obtained results with previous results, but there are no good explanations. Please explain each result.

L283-284: what is the relation between this study and this explanation?

Author Response

Communication: The effect of dietary humic substances on cellular immunity  and blood characteristics in piglets

Reviewer 2:

The study by Bujňák et al. assessed the effects of dietary supplementation of HS (0.5%) on some cellular immune indicators in pigs. The paper is well-written and exciting. Some comments should be considered before further processing

Abstract: please define all abbreviations first mentioned.

Why did the authors use only one supplement level (0.5%)? They should assess different graded levels to investigate the supplementation level that gives the best results finally.

Please avoid using “humic substances” and replace them with the exact supplement used.

Information about the number of replications, rearing, and management of pigs during the experiment should be provided.

L116-121: please mention in brief the methodology of analysis of these parameters.

Please insert the p-value in the tables (results)

The discussion only compares the obtained results with previous results, but there are no good explanations. Please explain each result.

L283-284: what is the relation between this study and this explanation?

Dear Reviewer,

Firstly, thank you for your time, willingness, and valuable comments, which helped us improve the overall quality of the revised manuscript. All accepted changes and modifications from all reviewers were marked up using "Track Changes" (MS Word) in the revised manuscript.

Yours sincerely,

team of authors

Specific comments

Reviewer 2

  • All the abbreviations first mentioned were defined. We used serum alkaline phosphatase instead of ALP in abbreviated form.
  • We understand your note about one supplement level (0.5%).

Allow us to clarify this:

  • Because this is a pilot study to assess the effect of 0.5% HS supplementation on immune parameters as well as biochemical and haematological variables in piglet blood, we prepared our manuscript as a communication or short note. 
  • The animals were housed in the accredited stables of the Department of Animal Nutrition and Husbandry of the University of Veterinary Medicine in Kosice at the time of this experiment; the capacity of these stables at the time allowed only two groups to be formed: the control group and the experimental group, each with 12 piglets.
  • Based on our previous results and experiences from poultry studies on the effect of humic substances on the immune status and immune response, we decided to verify this effect in piglets.
    • Mudroňová, D.; Karaffová, V.; Semjon, B.; Naď, P.; Koščová, J.; Bartkovský, M.; Makiš, A.; Bujňák, L.; Nagy, J.; Mojžišová, J.; Marcinčák, S. Effects of dietary supplementation of humic substances on production parameters, immune status and gut microbiota of laying hens. Agriculture, 202111, 744. DOI: 10.3390/agriculture11080744
    • Mudroňová, D.; Karaffová, V.; Pešulová, T.; Koščová, J.; Maruščáková, I.C.; Bartkovský, M.; Marcinčáková, D.; Ševčíková, Z.; Marcinčák, S. The effect of humic substances on gut microbiota and immune response of broilers. Food Agric. Immunol. 202031, 137–149. DOI: 10.1080/09540105.2019.1707780
  • At the moment, we are testing other concentrations and other properties (e.g. particle size) of the humic preparation in piglets.

  • In the part Materials and Methods (2.1.), we present information about the commercial name of the HS supplement (HUMAC® Natur AFM; Humac, Ltd., Košice, Slovakia) used in our study. We decided to use HS according to the recommendations of other reviewers and also because instead of the exact name of the supplement used, we used HS "humic substances" also in our previous articles as well (see below).
    • Semjon, B.; Bartkovský, M.; Marcinčáková, D.; Klempová, T.; Bujňák, L.; Hudák, M.; Jaďuttová, I.; Čertík, M.; Marcinčák, S. Effect of Solid-State Fermented Wheat Bran Supplemented with Agrimony Extract on Growth Performance, Fatty Acid Profile, and Meat Quality of Broiler Chickens. Animals202010, 942.
    • Hudák, M.; Semjon, B.; Marcinčáková, D.; Bujňák, L.; Naď, P.; Koréneková, B.; Nagy, J.; Bartkovský, M.; Marcinčák, S. Effect of Broilers Chicken Diet Supplementation with Natural and Acidified Humic Substances on Quality of Produced Breast Meat. Animals202111, 1087.
    • Mudroňová, D.; Karaffová, V.; Semjon, B.; Naď, P.; Koščová, J.; Bartkovský, M.; Makiš, A.; Bujňák, L.; Nagy, J.; Mojžišová, J.; Marcinčák, S. Effects of Dietary Supplementation of Humic Substances on Production Parameters, Immune Status and Gut Microbiota of Laying Hens. Agriculture202111, 744.

  • Information about the number of replications, rearing, and management of pigs during the experiment should be provided

  • The animals were housed in accredited stables.
  • Zoohygienic conditions: the stable's average temperature was 20.2 1.5 oC, and the relative humidity was 68.5 4.8%.
  • A total of 24 crossbred (Slovakian White x Landrace) 35 day-old piglets were divided into two groups (each with n = 12; 50% male and 50% female in both groups);
  • The pigs were fed twice per day with complete feed mixtures and were allowed ad libitum access to water throughout the experiment. The diets were formulated to meet or exceed the nutritional requirements recommended.

  • L116-121: please mention in brief the methodology of analysis of these parameters.

  • Serum biochemical parameter analysis was performed in an externally certified laboratory using a fully automatic analyzer.

  • Please insert the p-value in the tables (results)

  • It has been modified in the manuscript.

  • The discussion only compares the obtained results with previous results, but there are no good explanations. Please explain each result.
    • According to academic editor suggestions, we prepared this manuscript as a short note or communication. He suggested that we emphasize that we meant to point out the effects of HS primarily on the immune system.
    • Since the exact immunostimulatory effects of HS have not yet been sufficiently explained, we focused primarily on possible explanations for their immunomodulatory effect in past studies.

e.g. “…According to Cetin et al. [23], the addition of humic acid in laying hens' diet results in significant increases in the lymphocyte counts because of increased production of IL-2 and the expression of IL-2 receptors on lymphocytes, which led to the enhancement of the activity of IL-2 producing cells.“ ...

… The immunomodulation of HS may theoretically consist of the formation of complexes of humates with saccharides. These complexes bind to the surface of T lymphocytes and NK cells and affect their function, including the production of cytokines, which further influence other cells of the immune system [27].“ ...

  • In our study, we did not monitor the level of cytokines and therefore referred to the findings of other studies. In ongoing studies, where we monitor other concentrations of humic substances to confirm their immunostimulating effect, we also monitor the aforementioned parameters (cytokines) to confirm the mechanism of their immunostimulating effect.

  • L283-284: what is the relation between this study and this explanation?

  • We accept your note. It has been modified in the manuscript.

Reviewer 3 Report

The effect of dietary humic substances on cellular immunity  and blood characteristics in piglets

Dear Authors,

article is interesting and describes possibility addition of humic substances to the diet of piglets and their effect cellular immunity and blood characteristics. Obtained results shows that application of humic substances don’t give significant differences in haematological and biochemical parameters in blood serum expect of ALT concentration, what wasn’t obvious during start of experiment.  Use of those substances could have also for cellular immunity. Manuscript is clear and there are not many elements to correct. Below I add some suggestions helpful during this process:

Line 21

SI units should be used: g·kg-1.

Line 102

Table 1.

Name compounds could be changed for components,

Salt is common name and in this case better is to use NaCl or sodium chloride (as a precise determination of this substance),

Amount of each component in % can be changed for g·kg-1 ,

Metabolizable energy must be changed for MJ·kg-1.

Line 103

In case of premix is better to present amount of vitamins first (330000 IU vit. A) and after that minerals.

Line 175

SEM or standard deviation?

Line 188

Units should be present in standardized form T·L-1 and mmol·L-1 and so on

Line 283

Unit: kat·L1

Line 308

The same like in line 283

Line 347

3-4 articles need to be added to references (from years 2021-2023)

Author Response

Communication: The effect of dietary humic substances on cellular immunity  and blood characteristics in piglets

Reviewer 3:

Dear Authors,

article is interesting and describes possibility addition of humic substances to the diet of piglets and their effect cellular immunity and blood characteristics. Obtained results shows that application of humic substances don’t give significant differences in haematological and biochemical parameters in blood serum expect of ALT concentration, what wasn’t obvious during start of experiment.  Use of those substances could have also for cellular immunity. Manuscript is clear and there are not many elements to correct. Below I add some suggestions helpful during this process.

Line 21

SI units should be used: g·kg-1.

Line 102

Table 1.

Name compounds could be changed for components,

Salt is common name and in this case better is to use NaCl or sodium chloride (as a precise determination of this substance),

Amount of each component in % can be changed for g·kg-1 ,

Metabolizable energy must be changed for MJ·kg-1.

Line 103

In case of premix is better to present amount of vitamins first (330000 IU vit. A) and after that minerals.

Line 175

SEM or standard deviation?

Line 188

Units should be present in standardized form T·L-1 and mmol·L-1 and so on

Line 283

Unit: kat·L1

Line 308

The same like in line 283

Line 347

3-4 articles need to be added to references (from years 2021-2023)

Dear Reviewer,

Firstly, thank you for your time, willingness, and valuable comments, which helped us improve the overall quality of the revised manuscript. All accepted changes and modifications from all reviewers were marked up using "Track Changes" (MS Word) in the revised manuscript.

Yours sincerely,

team of authors

Specific comments

Reviewer 3

  • Your suggestions have been incorporated and modified in the manuscript.
  • The results obtained in this manuscript were expressed as "mean±SEM" and not as "mean±SD"? SEM stands for Standard Error of Mean.
  • Regarding references, two new references were added to the list and two others were removed.

Round 2

Reviewer 1 Report

.

Reviewer 2 Report

Thanks for the revision. No further comments.